:ͼ: PLOS | ONE

# Severe childhood anemia and emergency blood transfusion in Gadarif Hospital, eastern Sudan

**Mohammed Ahmed A. Ahmed[1], Abdullah Al-Nafeesah[2], Osama Al-Wutayd[3], Hyder M. Mahgoub[4], Ishag Adam** [ORCID][5]*

**1** Faculty of Medicine, Gadarif University, Gadarif, Sudan, **2** Department of Paediatrics, Unaizah College of Medicine, Qassim University, Unaizah, Kingdom of Saudi Arabia, **3** Department of Family and Community Medicine, Unaizah College of Medicine, Qassim University, Unaizah, Kingdom of Saudi Arabia, **4** New Halfa Hospital, New Halfa, Sudan, **5** Department of Obstetrics and Gynecology, Unaizah College of Medicine, Qassim University, Unaizah, Kingdom of Saudi Arabia

* ishagadam@hotmail.com

**Data Availability Statement:** All relevant data are within the paper and its Supporting Information files.

## Abstract

### Background

Anemia is a major cause of global morbidity and mortality, particularly among children. Management of anemia depends on causes and severity of anemia. However blood transfusion is a lifesaving intervention in severe and life-threatening anemia. There are no published data on blood transfusion for anemia in Sudan.

### Methods

A descriptive study was conducted in Gadarif Hospital in eastern Sudan during 1 August, 2017 to 31 March, 2018. Consecutive children who presented at the emergency room with an indication for blood transfusion were enrolled in the study. A detailed history was gathered from all patients. Physical examinations, including vital signs, were performed. The World Health Organization guidelines for blood transfusion were followed.

### Results

During the study period, a total of 1800 children were admitted to the emergency pediatric ward in Gadarif Hospital and were assessed for anemia, 513 (28.5%) were anemic and 141 (7.8%) had severe anemia. Three hundred anemic children received blood transfusion. The median (interquartile) of the age of the 300 children who received blood transfusion was 4.2 4.2(2.0–9.0) years. A total of 148 (49.3%) of the children were boys and 151 (50.3%) were younger than 5 years. The diagnoses associated with the order for blood transfusion were sickle cell disease (129, 43.0%), active bleeding (58, 19.3%), malaria (50, 16.7%), visceral leishmaniasis (25, 8.3%), severe acute malnutrition (16, 5.30%), snake bite (11, 3.7%), sepsis (5, 1.7%), and others. Two hundred eighty-five (95.0%) children improved, nine children were discharged against medical advice, and six (2.0%) children died.

**Funding:** The authors received no specific funding for this work.

**Competing interests:** The authors have declared that no competing interests exist.

## Conclusion

There is a high burden of anemia in eastern Sudan. Sickle cell disease, malaria, and visceral leishmaniasis are the main causes of anemia in this region. Further research on blood transfusion is needed.

## Introduction

Anemia in children is a large health problem and it is the main cause of global morbidity and mortality [1]. Severe anemia in sub-Saharan African children is a major cause of admission to the hospital, as well as a leading cause of mortality [2]. There are a variety of causes that lead to severe anemia, such as severe malaria, bacterial infection, sickle cell disease, and malnutrition [3–7]. Management of anemia depends on the causes and severity of anemia. However, blood transfusion is a lifesaving intervention in severe and life-threatening anemia [8].

Anemia is common in low and middle income countries where availability of blood is, sometimes, beyond reach with delays in acquisition or administration, a major factor of mortality in children with severe anemia [9–11]. The World Health Organization (WHO) has provided guidelines for informing about clinical decisions on transfusion associated with hemoglobin levels[12]. Unfortunately, there is poor adherence to current WHO transfusion guidelines in African countries [2].

Severe anemia secondary to infections, such as malaria, Leishmania species, and Shistosoma species, is a large health problem in Sudanese children [13–15]. While there are published data on blood transfusion in children in other African countries [10,11,16–18], there are no published data on blood transfusion in Sudan. Therefore, the current study was conducted to determine the rate of anemic children with WHO indications for blood transfusion, and to describe the indications for blood transfusion the time to blood transfusion and outcome of blood transfusion, in children in Gadatif Hospital in eastern Sudan.

## Methods

We conducted a cross-sectional study in Gadarif Hospital in eastern Sudan during 1 August, 2017 to 31 March 2018. Gadarif is 400 km from the capital Khartoum on the Ethiopian and Eritrean borders. Gadarif city is located between latitudes 14˚ and 16˚ north and longitudes 33˚ and 36˚ east. This city is at an altitude of 496 m above sea level, with a population of 1,727,401 inhabitants. Gadarif Hospital is a 400-bed tertiary care facility that serves as a referral center for Gadarif State. The average patient turnover at this hospital is 150–200 patients per day. The emergency pediatric ward is staffed with three consultants, four specialists, and 14 medical doctors (registrars and residents).

Consecutive children (aged <18 years) who presented at the Outpatient Department and the Children's Emergency Room with indication for blood transfusion were enrolled in the study. After the children's parent(s) or guardians signed an informed consent form, a detailed history was gathered from all patients or their guardian. Physical examinations, including vital signs, were performed according to standardized procedures at the discretion of the treating doctor as part of standard care. Samples of urine and stool were collected, and they were investigated for schistosomal infection as described before [19,20]. Blood was withdrawn from the median antecubital vein and examined for malaria using thick and thin blood films which were stained with 10% Giemsa and read by expert microscopist. Hemoglobin estimation was

done as part of the complete hemogram via automated blood-cell analyzer machine (Sysmex Hematology Analyzer; Sysmex, Kakogawa, Japan).

According to WHO, a child between 6 and 59 months is defined as "anemic" when the hemoglobin level is below 11.5 g/dl ("severely anemic" hemoglobin <7 g/dl). Between 5 and 11 years "anemia" is present with hemoglobin below 11.0 g/dl; between 12 and 14 years hemoglobin below 12.0 g/dl and from 15 years on hemoglobin below 13.0 g/dl. Except for the first age class, the definition of "severe anemia" is restricted to patients with hemoglobin level below 8.0 g/d [21].

Hemoglobin electrophoresis was performed on blood samples from patients who were clinically suspected to have sickle cell disease (after initial sickling test). Electrophoresis (electrophoretic equipment model MUPID-EXU, Japan) was done for those who were not diagnosed before; those who were already diagnosed were labeled as sickle cell disease.

The WHO guidelines for blood transfusion were followed [12,22] and these are as follows: hemoglobin levels <4 g/dl or hemoglobin levels of 4–7 g/dl plus shock, clinically detectable dehydration, impaired consciousness, respiratory acidosis as shown by deep breathing, heart failure, and/or more than 20% of red blood cells parasitized by malaria parasites; or hemoglobin levels >4 g/dl with continuing bleeding. Children were transfused with 20 mL/kg whole blood or 10 mL/kg packed cells, which were provided for no longer than 4 hours. Furosemide (1 mg/kg intravenously) was used at the beginning of blood transfusion for children with clinical signs of pulmonary edema. Pulmonary edema was clinically determined if the patient had shortness of breathing, increased respiratory rate with coarse crackles at the lung bases and radiologically as defined by the presence of Kerley B lines in the anteroposterior chest view. Severe acute malnutrition was diagnosed following the WHO guidelines for malnutrition. A child was considered as severe acute malnutrition if weight-for-height z score was <-2 SD for age and sex or presence of bilateral lower limb edema [23,24]. The diagnosis of visceral leishmaniasis was confirmed by the visualization of the amastigote form of the parasite by microscopic examination of aspirates from or bone marrow using Giemsa-stain.

The definition of sepsis was considered as "life threatening organ dysfunction caused by a dysregulated host response to infection [25]. Any severe adverse events were reviewed by the treating clinician. The time taken from order of blood to actual transfusion was calculated from the time when transfusion was prescribed to the time of actual transfusion. The outcome following blood transfusion was classified as survived (recovered), died, or left the hospital against medical advice. Other managements/treatments, such as antimalarials, antibiotics, and treatment for Leishmania species, were provided according to the diagnosis and the National guidelines [13,14].

A sample size of 300 participants was calculated as the sample size in a finite population using the formula N = Z$^2$ P (1-P)/e$^2$ (N = sample size, Z = level of confidence, P = baseline level of the selected indicator and e = margin of error. P was estimated at 0.50). This was determined by the prevalence (48.9%) of severe anemia in children who were previously admitted with visceral leishmaniasis in the same hospital [13]. The calculated sample size had 80% power and precision of 5% at $\alpha$ = 0.05. We assumed that 10% of the children might have incomplete data.

## Ethics

Ethical approval was received from the Ethics Committee at the Faculty of Medicine, Gadarif University, Sudan (reference number: 2016/08). Written informed consent was collected from each participant's parents (or guardian) before taking part in the research.

## Statistical analysis

Data were entered into a computer using SPSS for Windows (version 20.0). Continuous data were checked for normality using Shapiro-Wilk test and they were presented as mean (Standard Deviation–SD) if they were normally distributed or median [interquartile range–IQR] if they were not normally distributed. Frequencies and proportions were calculated.

## Results

During the study period between, a total of 1800 children were admitted to the emergency pediatric ward in Gadarif Hospital. Of the 1800 children, 513 (28.5%) were anemic and 141 (7.8%) had severe anemia.

Three hundred anemic children were admitted to the emergency pediatric ward and received blood transfusion.

The median (interquartile) of the age of the study population (300 children who received blood transfusion) was 4.2 4.2(2.0 –9.0) years and 148 (49.3%) of them were boys. A total of 151 (50.3%) were children aged younger than 5 years. The median (interquartile) of their hemoglobin was 5.0 (3.9–6.3)g/dl (Table 1).

The distribution of hemoglobin level of the transfused children is shown in Fig 1.

While fever ($\geq$37.5˚C) (262, 87.3%) and difficulty in breathing (193, 64.3%) were the predominant symptoms, pallor (274, 91.3%), and hepatomegaly (215, 71.7%) were the main signs that were detected in children who received blood transfusion Table 2.

The diagnosis associated with order of blood transfusion is shown in Fig 2.

The vast majority (115/129, 89.1%) of the sickle cell disease were diagnosed before. Hemoglobin electrophoresis was performed to 14/129 children. Of the 14 children, eight (57.1%), five (35.7%) and one child (7.1%) had hemoglobin SS, hemoglobin AS and hemoglobin AC, respectively. There was no case of intestinal or urinary schistosomiasis.

Seventy-six (25.3%) children were transfused because had hemoglobin levels <4 g/dl and 204 (68.0%) children had hemoglobin levels of 4–7 g/dl. Of 204 children who had hemoglobin levels of 4–7 g/dl and were transfused; 110 (53.9%) had heart failure, 38 (18.6%) had > 20% of red blood cells parasitized by malaria parasites, 32 (15.7%) had continuing bleeding, 9(4.4%) had combined/ others reasons, 8(3.9%) had dehydration and 7(3.4%) had impaired level of consciousness.

**Table 1. Patient's characteristics and their outcomes.**

|  | Number (%) | Age, median (interquartile) | Hemoglobin, median (interquartile) |
|---|---|---|---|
| **Admitted** | 1800(100.0) | 5.2(2.0–10.5) | 12.8(7.2–13.9) |
| **Non anemic** | 1287 (71.5) | 5.3(2.1–10.3) | 12.9 (11.1–14.1) |
| **Anemic** | 513 (28.5) | 5.1(2.0–10.4) | 9.8(9.5–10.6) |
| Mild–moderate | 372 (20.6) | 5.0(2.1–9.8) | 8.1(6.8–9.8) |
| Severe | 141 (7.8) | 5.2(2.0–10.5) | 3.2(2.8–3.6) |
| **Received transfusion** | 300 (13.6) | 4.2(2.0 –9.0) | 5.0(3.9–6.3) |
| Age < 5 years | 151 (50.3) | 2.0(1.0–3.0) | 5.0(4.0–6.2) |
| Hb < 4.0 g/dL | 76 (25.3) | 6.0(2.0–11.7) | 3.1(2.6–3.7) |
| Hb $\geq$ 4–$\leq$ 7 g/dL | 204 (74.3) | 4.0(2.0–8.7) | 5.4(4.8–6.8) |
| Improved | 285 (95.0) | 4.0(2.0–9.0) | 5.0(3.9–6.3) |
| Discharged against medical advice | 9 (3.0) | 10(7.0–11.0) | 6.0(4.4–7.0) |
| Died | 6 (2.0) | 2.5(1.0–10.0) | 5.0(3.3–7.5) |

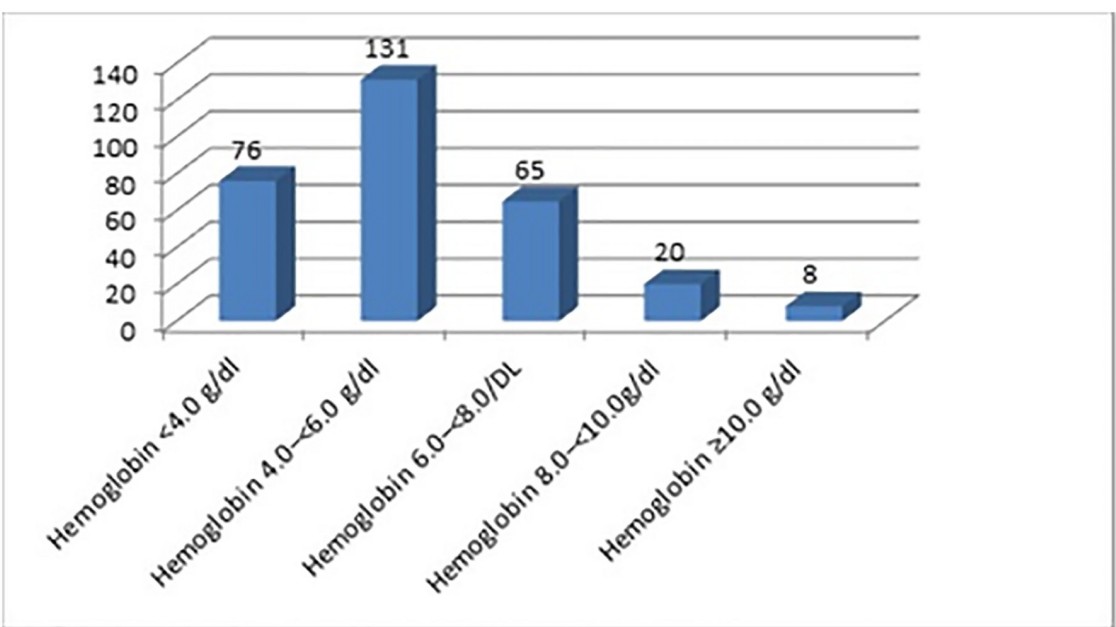

**Fig 1. Distribution of hemoglobin level of the transfused children.**

The median (interquartile range) request-to-issue time for blood was 6 (3–12) hours.

Five (1.6%) children had a "probable" blood transfusion reaction in the form of fever (1˚C above the baseline temperature) and all of them had an urticarial rash within 3 hours of starting blood transfusion. Thereafter, the transfusion was stopped and these children received intravenous hydrocortisone. All of the five children completely recovered.

A total of 285 (95.0%) children improved, nine (3.0%) children were discharged against medical advice, and six (2.0%) children died. Four of the children who died were girls, and two of them had severe malaria, two had malnutrition, and one had sickle cell disease. Two of the

**Table 2. Symptoms and signs in children who received blood transfusion in Gadarif Hospital, Sudan.**

| Variables | Frequency | Proportion (%) |
|---|---|---|
| *Symptoms* | | |
| Difficulty in breathing | 193 | 64.3 |
| Fever | 262 | 87.3 |
| Weakness | 78 | 26.0 |
| Vomiting | 46 | 15.3 |
| Cough | 83 | 27.7 |
| Diarrhea | 30 | 10.0 |
| Active bleeding | 58 | 19.3 |
| *Signs* | | |
| Pallor | 274 | 91.3 |
| Hepatomegaly | 215 | 71.7 |
| Splenomegaly | 83 | 277 |
| Jaundice | 48 | 16.0 |
| Pedal edema | 23 | 7.7 |

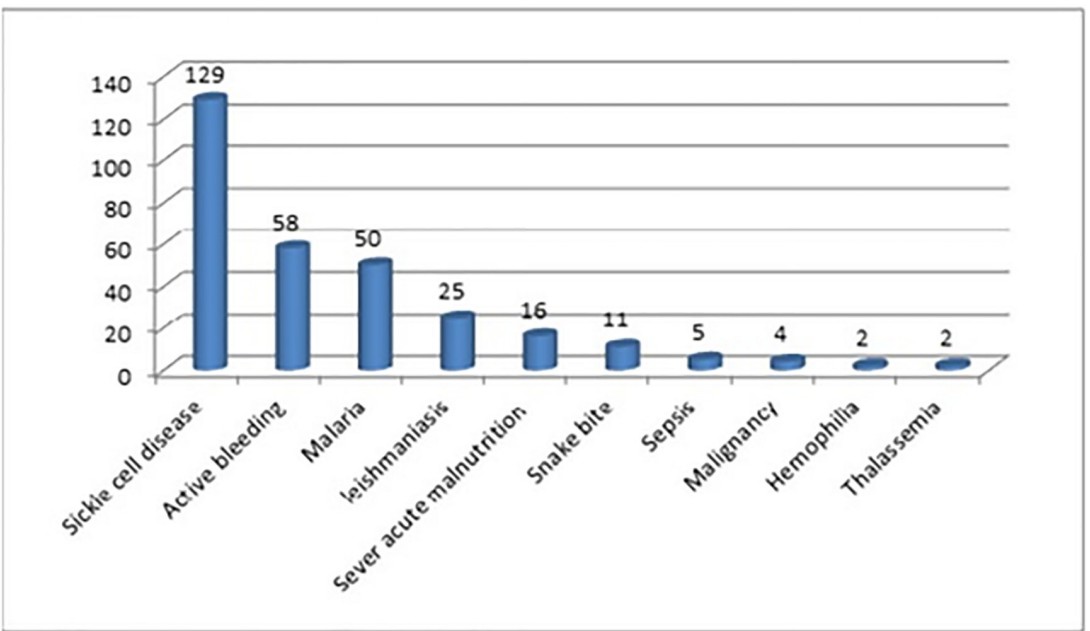

**Fig 2. The diagnoses associated with blood transfusion.**

children who died had hemoglobin levels <4 g/dl. The mean (range) time taken for blood transfusion in the six children was 4.3 (2–6) hours.

## Discussion

To the best of our knowledge, this is the first report on blood transfusion in Sudanese children. This paper is to provide information on the rate of anemic children who received blood transfusion and to describe the indications, outcome, time needed to transfuse and mortality following transfusion in eastern Sudan. The main findings of the current study were as follows. The prevalence of anemia in children who presented to the hospital was 28.5% and 7.8% had severe anemia. Sickle cell disease and malaria were the main diagnoses associated with anemia requiring blood transfusion.

The prevalence of anemia in our study is slightly higher than the estimated prevalence of anemia globally(24.8%) [1]. In a recent meta-analysis the global anemia prevalence in 2010 was 32.9% [26]. However, our rate of anemia is much lower than that among children in other African countries (e.g., two thirds of Tanzanian children who are tested in the Emergency Department have anemia) [10]. The prevalence of severe anemia in our study is similar to the reported prevalence (9.7%) of severe anemia among children in Nigeria [27]. However, the prevalence of severe anemia in our study is much lower than that in other African countries (12% in Kenya, 41% in Uganda, and 42% in Tanzania) [2,10]. The difference in the prevalence of anemia in our study and other studies could be explained by a difference in the enrolled participants. Children of all age groups were enrolled in our study, while in most of the other studies, children aged younger than 5 years were enrolled. Furthermore, eastern Sudan is characterized by unstable transmission of malaria [28], while most of the other African countries have stable transmission of malaria. Previous reports have shown that severe malaria and severe anemia are the most common morbidities in eastern Sudan [6,29]. Severe anemia in Sudanese children is area/region-specific (e.g., Shistosomal infection). Shistosomal infection

was reported as the main cause of severe anemia of children in irrigated areas in eastern Sudan [7]), and was not observed in this study.

Our study showed that the diagnoses associated with the order for blood transfusion were sickle cell disease (129, 43.0%), active bleeding (58, 19.3%), malaria (50, 16.7%), visceral leishmaniasis (25, 8.3%), and severe acute malnutrition (16, 5.30%). In Tanzania, severe malaria (20.1%), sickle cell disease (18.3%), septicemia (7.6%), and severe malnutrition (5.4%) are the major causes of admission to the emergency pediatric ward [10]. Similarly, fever (26%), vomiting (8.3%), cough (8.1%), general body malaise (7.6%), difficulty in breathing (6.5%), and diarrhea (5.7%) are the predominant complaints in children who are admitted to the Emergency Pediatric Department in Tanzania [10]. In neighboring Kenya, blood transfusion is mostly used to treat severe malaria-associated anemia and sickle cell disease, and to replace massive blood loss [8,11]. In Ghana, malaria was reported as the main factor for requirement for blood transfusions in children [18]. However, septicemia (19, 13.6%), sickle cell disease (13, 9.3%), and malnutrition (10, 7.1%) are the most common indications for blood transfusion in Nigerian children [27].

In the current study 91% of the children had pallor, 87% of them presented with fever and 64% presented with difficulty in breathing, yet only 141 (47%) were diagnosed as severe anemia. Perhaps the symptoms might not be a useful indicator for anemia in this setting.

Approximately two fifths of the transfused children in the current study had sickle cell disease. A recent report showed that one in every 123 children in eastern and western Sudan is at risk of having sickle cell disease [30]. The (Misseriya tribes) ethnicity and the consanguineous marriages were the main explanations of the high prevalence (24.9%) of carriers of HbS allele (HbAS) in certain areas of Sudan [31]. A much lower rate (9.3%) of sickle disease has been reported in transfused Nigerian children [27]. Notably, some African countries, such as Cameroon, Gabon, Ghana, and Nigeria, have a high incidence of sickle cell disease (20%–30%) and it is much higher in Uganda (45%) [32].

In the current study, the median (interquartile range) request-to-issue time for blood was 6 (11–28) hours, which is unacceptably prolonged for urgent requests. Nabwera et al. reported a median request-to-issue time of between 3.6 and 5.4 hours in a pediatric referral hospital in Kenya [17]. In Tanzania, the median time to transfusion was 7.8 hours in anemic children who were transfused in a urban tertiary hospital [10]. Therefore, our study showed a much longer time to transfusion than the recommended guidelines from optimum healthcare systems. These guidelines suggest availability of uncross-matched blood within 10 minutes and availability of group-specific blood within 30 minutes [33]. A delay in transfusing children is associated with adverse outcomes [2,11]. Interestingly, it has recently been shown that there was no significant difference in clinical outcomes between the children who received immediate transfusion and those who did not (no immediate transfusion, control group)[34].

The "probable" blood transfusion reaction in our study was higher than that reported by Kiguli et al. (6/1,387, 0.4%) in other African countries [2]. We found that 285 (95.0%) children improved, nine (3.0%) children were discharged against medical advice, and six (2.0%) children died. In Nigeria, 117 (83.6%) children with anemia recovered, while four (2.8%) left the hospital against medical advice and 19 died [27]. In our study, the overall mortality rate in transfused children was 6/300 (2.0%). This rate is lower than that reported among anemic children in Tanzania (12.1%) [10] and in Nigeria (13.6%) [27]. However, the mortality rate in our study is similar to that (4%) reported in in Uganda, Kenya, and Tanzania in children with severe febrile illness [2]. Mortality in children with severe anemia is high when hemoglobin levels are <4 g/dL or if there is associated respiratory distress [35]. This difference among studies could be explained by differences in participants (age of the enrolled children), as well as differences in the definition of severe anemia [2].

## Limitations of the study

Some factors such HIV was not investigated. HIV has been reported as the leading cause of severe anemia in other African countries [4]. However, a recent meta-analysis showed a low (1.0%) prevalence of HIV in Sudan [36].

## Conclusion

There is a high burden of anemia in eastern Sudan. Sickle cell disease, malaria, and visceral leishmaniasis are the main causes of anemia in this region. Future reach on blood transfusion is required.

## Supporting information

**S1 Table. Table of the raw data included in this paper.**
(XLSX)

## Acknowledgments

We thank Ellen Knapp, PhD, from Edanz Group (www.edanzediting.com/ac) for editing a draft of this manuscript.

## Author Contributions

**Conceptualization:** Mohammed Ahmed A. Ahmed, Osama Al-Wutayd, Hyder M. Mahgoub, Ishag Adam.

**Data curation:** Osama Al-Wutayd, Hyder M. Mahgoub.

**Formal analysis:** Abdullah Al-Nafeesah, Osama Al-Wutayd, Hyder M. Mahgoub, Ishag Adam.

**Methodology:** Mohammed Ahmed A. Ahmed, Abdullah Al-Nafeesah.

**Validation:** Ishag Adam.

**Writing – original draft:** Mohammed Ahmed A. Ahmed, Abdullah Al-Nafeesah, Osama Al-Wutayd, Hyder M. Mahgoub.

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
