## [Decision Letter · Decision Letter 0]

2 Sep 2019

PONE-D-19-15934

Severe childhood anemia and emergency blood transfusion in Gadarif Hospital, eastern Sudan

PLOS ONE

Dear Professor Adam,

Thank you for submitting your manuscript to PLOS ONE. After careful consideration, we feel that it has merit but does not fully meet PLOS ONE’s publication criteria as it currently stands. Therefore, we invite you to submit a revised version of the manuscript that addresses the points raised during the review process.

The manuscript has been assessed by two reviewers; their comments are available below.

The reviewers have raised some major concerns that need attention in a revision. The reviewers raise that the manuscript should provide a clear definition for anemia, as well as definitions for other conditions measured. The reviewers request additional details on methodological aspects of the study and note concerns about the possibility of selection bias in the population. The reviewers also note that further statistical analyses must be carried out and the presentations of the data improved.

Could you please revise the manuscript to carefully address the concerns raised by the reviewers?

We would appreciate receiving your revised manuscript by Oct 15 2019 11:59PM. Please include the following items when submitting your revised manuscript:

We look forward to receiving your revised manuscript.

Kind regards,

Iratxe Puebla

Senior Managing Editor, PLOS ONE

Journal Requirements:

1. Thank you for including your funding statement; "None received"

Please provide an amended Funding Statement that declares *all* the funding or sources of support received during this specific study (whether external or internal to your organization) as detailed online in our guide for authors at http://journals.plos.org/plosone/s/submit-now.  

Please state what role the funders took in the study.  If any authors received a salary from any of your funders, please state which authors and which funder. If the funders had no role, please state: "The funders had no role in study design, data collection and analysis, decision to publish, or preparation of the manuscript."

Reviewers' comments:

Reviewer's Responses to Questions

**Comments to the Author**

1. Is the manuscript technically sound, and do the data support the conclusions?

Reviewer #1: No

Reviewer #2: Yes

2. Has the statistical analysis been performed appropriately and rigorously? 

Reviewer #1: No

Reviewer #2: N/A

3. Have the authors made all data underlying the findings in their manuscript fully available?

Reviewer #1: No

Reviewer #2: No

4. Is the manuscript presented in an intelligible fashion and written in standard English?

Reviewer #1: Yes

Reviewer #2: Yes

5. Review Comments to the Author

Reviewer #1: Reviewer’s report for manuscript # PONE-D-19-15934

Summary: This was a cross sectional study conducted at a tertiary hospital in Eastern Sudan, in which the investigators attempted to determine the clinical use of blood among children receiving urgent transfusion. They found that approximately 56.5% (300/531) of children with anemia presenting to the emergency unit were transfused. Sickle cell disease (43%), acute bleeding (19.3%) and malaria (16.7%) were the leading diagnostic categories for which urgent blood transfusions were given.

Overall impression

The authors conducted a descriptive study, but methods of which lack details. There is a major concern for selection bias, since hemoglobin electrophoresis was only done among some patients, and yet sickle cell disease turned out to be the leading diagnostic category for transfusion. The data analysis seems incomplete (for the type of study design), since they could not even establish the factors associated with the main outcome variables. Consequently, the claims/answers to the current study are not supported by credible results. The authors make several speculations, such as about HIV, Schistosomiasis, and Sickle cell disease with no supporting data. Similarly, the results of some stated objectives are either not available or cancelled; probably for tests that were never performed. Results presentation and discussion deserve major revisions.

Specific issues, in details

Major issues

a) Introduction:

1. The objective(s) of the study, as stated in line 92-96, are not clear. Please revise, and clarify on the following:

i) Was the aim to ‘assess the rate’ of anemic children who….or to the ‘determine the proportion’ of anemic children who receive transfusion….?

ii) Did the authors examine (the indication, patterns and outcomes…), or describe?

b) Methods:

1. The authors need to clarify on the type of study design (line 98). Was this a cross-sectional study?

2. Generally, the methods lack details. By providing details of the scientific methods used in a clinical study, the researcher assures the global community that – another person, using the same tools may reproduce similar findings. Please explain and clarify on the following:

i) Samples of urine and stool were taken off (line 113); Provide details of the specific tests that were performed, and the machines used.

ii) Line 114-115, mentions that blood was examined for malaria. Please provide details; which films: thick films, thin films, or both? Which stains were used? Who read them?

iii) The diagnosis of anemia is very crucial in this study (refer to line 116). How was hemoglobin determined? Please provide details (e.g which machine was used?)

iv) Hemoglobin electrophoresis (refer to line 116). Provide details of the machine and techniques used.

v) Hemoglobin electrophoresis (refer to line 116). The authors say this was done “if required in some cases”! This is a major concern for selection bias, if such a test that confirms sickle cell status was only performed among some patients. Remarkably sickle cell disease turned out to be the leading diagnostic category for transfusion, yet some children were NOT tested! Please explain.

vi) In line 123, the authors state that transfusions were given for longer than 4 hours. However, transfusions are generally given for a period 2-4hrs. The practice of transfusion for longer than 4hrs is not safe and increases the risk for bacterial contamination. Is this the standard practice at Gadarif hospital? Please Explain.

vii) Similarly, explain how the diagnosis /suspicion of pulmonary edema was made. There is a risk for irrational use of frusemide.

viii) The text in line 126 is confusing. Do you mean “the time taken from order, to actual transfusion”? This may not be referred to as ‘time of blood transfusion’. Please modify.

ix) There are many other details missing, that should be mentioned such as:

x) How was the diagnosis of acute malnutrition made? Provide details.

xi) How was the diagnosis of visceral leishmaniasis made? Provide details.

xii) How was the diagnosis of sepsis made?

3. Statistical analysis:

i) Generally, additional statistical analysis needs to be done, such as univariate and bivariable analysis - to establish factors associated with the main outcomes such as survival, etc.

ii) Line 145: were all the numerical data symmetrical, to justify the use of Mean (SD)? If not, use median for asymmetrical data. Please check and revise.

4. For this study, the authors need to specify their operational definitions. For example;

i) What defined malaria?

ii) What defined sickle cell disease?

iii) What defined acute malnutrition?

iv) What defined acute bleeding?..etc.

c) Results:

1. Generally when presenting results, Table 1; which is the table of baseline characteristics comes up-front. These characteristics may include; gender, age, duration of illness, relationship with primary caregiver, occupation of caregiver, ethnic group/tribe etc. This table is missing.

2. Present the results systematically; objective by objective.

3. Results of some stated objectives are missing, such as; ‘pattern’ and ‘time taken from order, to actual transfusion’. Provide them.

4. Samples of urine and stool were taken off. Provide the results of these tests.

5. Hemoglobin electrophoresis was done. Provide detailed results, including the proportion with SS, AS etc.

6. Figure 1(line 154-166): The flow chart:

Consider revising this flow chart, such as below (only a guide);

.....Chart is attached, separate file,

7. The authors should account for each study participant. For example, 141/300 transfused children were categorized as severe anemia. What was the diagnosis among the 159 with Hb>5 but got transfused? Please explain and modify.

8. These 159, deserve a thorough discussion (in the discussion section).

9. This account above is important, because the description offered in line, 170 -173; does not seem to tie-up with what we normally see clinically: That 91% had pallor, 87% with fever and 64% with breathlessness, yet only 141 (47%) were diagnosed as severe anemia. Please clarify.

10. Line 181: Figure 2: This is a duplication of presentation of findings. Generally it is not necessary to show both the text (177-180) and figure. You choose either.

If choose text, then one writes...’the diagnosis associated with order of blood transfusion is shown in figure 2, below:’

11. Line 189: For the five that developed adverse events, provide more details. E.g. After stopping, what happened to them? Full recovery? Where these 5 among the the 6 deaths? or the 9 that discharged self..

12. Fully account for each participant; such as in # c(11), and c(6) - c(8) above.

13. Line 190: The phrase; “A total of 285 children improved…” is left hanging. So what factors were associated with recovery? This entire paragraph (190-194) does not make sense, without additional statistical analysis, as pointed out in c(3) above. Please revise.

d) Discussion:

1. What is the burden of Sickle cell disease in Sudan? In line 236, they mention one in 123 children. Do the authors believe this figure/prevalence? ….especially in light of their current finding that show; 129/300 (43%) of transfused children in that same setting have sickle cell disease? Please explain, and discuss.

2. This section is generally poorly written (flow), with lots of repetitions. Suggested flow may include the following areas (paragraphs):

i) First paragraph – what was the aim of the study/ what was your question?

ii) What did you find (the answer to your question)

iii) A brief explanation of the findings, in comparison/contrast with other studies.

iv) What is new in the current study?

v) The strength of your study.

vi) Limitations

vii) Conclusions /or and recommendations.

e) Conclusion:

Generally, the conclusion shall need to be revised, once most of the above revisions are finalized, especially those related to additional statistical analysis.

Minor issues

a) Abstract:

1. Were all the 1,800 children evaluated for anemia? Please clarify.

2. The text in line, 46-47 is confusion. “The diagnoses associated with the order for anemia…” Do clinicians order for anemia? Consider revising text, such as; “The diagnoses associated with the order for blood transfusion were…..” Do the same elsewhere, such as line 177.

b) Introduction:

1. The text in line 72-73 lack clarity. Consider revising text, such as; ‘Childhood anemia is a major cause of morbidity and mortality globally.

c) Methods:

1. Line 99, add ‘city’ to ‘Gadarif’ so that text reads; Gadarif city is 400km…..

2. Sample size estimation: Specify the formula used; such as Kish, Leslie for cross-sectional studies.

d) Results,

1. Line 170: The authors need to use terminologies correctly and in context. By ‘breathlessness’ do the authors mean, respiratory distress or difficulty in breathing? Please clarify.

2. Similarly, Line 171: Palpitation is a rare symptom among children, especially in <5 years who often don't report symptoms. Remember 151 (50.3%) were <5 years! In addition, in table 1; palpitation is grouped as a sign! This is confusing. Please revise.

3. Line 184: revise the phrase, such as; Median time, from order to time of blood issue (or better start of transfusion) – as appropriate.

4. Line 186: For fever, generally we consider an increment of >1°C, from baseline. Was this the case? Please clarify.

5. Line 187 (Adverse events are very important): Better specify; of the 5, how many got fever, and how many got urticarial rash?

e) Discussion:

1. Line 201: The authors state that, “the prevalence of anemia in our study was slightly higher than…”. Why? Please discuss. But, first clarify on how the 531(diagnosed with anemia) were got from the 1,800...to exclude selection bias. In other words, were all the 1,800 admitted children evaluated for anemia? Please clarify.

2. The phrase in line 215 conveys no meaning, especially with regard to what is put in parenthesis or (....). Please revise.

3. Line 216-217, about schistosomiasis: Are you sure? Please provide results of stool and urine, showing how many did not have. These results are missing.

4. Line 218-219; how did you know? When HIV was not tested. No mention of HIV testing in the methods or results.

5. Line 252-253: This phrase is not accurate; 0.4% in Kiguli et al study is not comparable to 1.6%! Please revise or explain.

f) Other comments: None.

Reviewer #2: PLONE-D-19-15943

Title

Severe childhood anemia and emergency blood transfusion in Gadarif Hospital, eastern Sudan

GENERAL COMMENTS

In this study the Authors assessed the prevalence of anemia in children admitted to their hospital in Sudan with an indication for blood transfusion through an eight months period. We subsequently examined a series of pre- and post- transfusion parameters (underlying disease, symptoms, signs, patients’ outcome and others).

The study, as the first performed in Sudan in this field, appears original and useful to better understand the real prevalence of anemia, its causes, the indications to blood transfusion, the transfusion behavior in a pediatric emergency department and the patients’ outcome.

However, I have some remarks and suggestions for changes.

First of all, I think that a clear statement (with an appropriate reference) must be included with respect to the definition of anemia used by the Authors, because this can affect the results.

According to WHO, a child between 6 and 59 months is defined as “anemic” when the hemoglobin level is below 11.5 g/dl (“severely anemic” below 7 g/dl). Between 5 and 11 years “anemia” is present with Hb below 11.0 g/dl; between 12 and 14 years below 12.0 g/dl and from 15 years on below 13.0 g/dl. Except for the first age class, the definition of “severe anemia” is restricted to patients with hemoglobin level below 8.0 g/dl.

See: World Health Organization. Haemoglobin concentrations for the diagnosis of anaemia and assessment of severity. WHO; 2011.

In the Results section of the abstract (line 42) the sentence was: “…513 (28.5%) were anemic (hemoglobin < 11 g/dl) and 141 (7.8%) had severe anemia” (hemoglobin level < 5 g/dl).” Please adjust this point.

In the statistical section the Authors declare that data are presented as mean (with standard deviation), but in line 184-185 they cite a “median (interquartile range)”. In line 194 the haemoglobin level should be expressed as median [IQR], especially since the word in brackets is “range”. Median and IQR are also cited in lines 242 and 244.

In my opinion the sentence in statistical section should be “Data are presented as mean (± Standard Deviation – SD) or median [interquartile range – IQR]

In the same sections the Authors should be described the methodology used to calculate the sample size (see line 132 – 135)

OTHER SPECIFIC COMMENTS

Introduction, Line 72 – 88

I would suggest a complete revision of the text, to make it more concise. The contents are appropriate, but there is a series of repetitions and redundancies.

Line 85

The reference n.12 (“Guidelines for informing about clinical decisions on transfusion associated with hemoglobin levels” ) doesn’t correspond to the WHO’s guidelines. In my opinion it could be better the reference n.20.

Line 123

Furosemide instead of frusemide

Line 135

A difference of 5% of what? Please specify.

Line 148 (Patients’ characteristics)

It is the only subheading in the “Results” section: the Authors may choose to erase it, to add other subheadings, if deemed appropriate.

Line 151

The question is always the same: is the definition of severe anemia arbitrarily chosen or reference-based? It is sufficient to declare it.

Figure 1. and lines 167-169

I suggest a table instead of a flowchart with the subsequent descriptive text.

The table is in the attachment named as “Suggested Table”

I would appreciate a bar chart with hemoglobin level classes at admission (e.g. < 4 / 4 - <5 / 5 - < 6 and so on) in the x axis and the prevalence in the y axis.

Line 194

See “statistic section” in the General comments

Discussion

Line 202

I suggest to add the following reference: Kassebaum NJ, Jasrasaria R, Naghavi M, Wulf SK, Johns N, et al. A systematicanalysis of global anemia burden from 1990 to 2010. Blood 2014(123):615–24. https://doi.org/10.1182/blood-2013-06-508325

Line 251

I suggest to add also the following reference and include a short discussion on its different results respect to the articles in reference 2 and 11. Maitland K, Kiguli S, Olupot-Olupot P, Engoru C, Mallewa M, Saramago Goncalves P, et al. Immediate Transfusion in African Children with Uncomplicated Severe Anemia. New England Journal of Medicine. 2019 Aug 1;381(5):407–19.

Conclusion

“Future reach on anemia is required” The phrase is not clear: please reword.

Lastly, it could be interesting to know the level of blood transfusion appropriateness. Are the WHO guidelines always followed? Did the Authors document cases of undertransfusion or non-transfusion in patients in need for homologous blood supply?

6. PLOS authors have the option to publish the peer review history of their article (what does this mean?). If published, this will include your full peer review and any attached files.

Reviewer #1: No

Reviewer #2: No

---

## [Author Response · Author response to Decision Letter 0]

20 Sep 2019

We would like to thank the editor and reviewers for their valuable comments on this manuscript. We greatly appreciate your time dealing with the manuscript and assessing the review comments, which enabled us to greatly improve the quality of our manuscript. Here is a point-by-point response to the reviewers’ comments and concerns

5. Review Comments to the Author

Summary: This was a cross sectional study conducted at a tertiary hospital in Eastern Sudan, in which the investigators attempted to determine the clinical use of blood among children receiving urgent transfusion. They found that approximately 56.5% (300/531) of children with anemia presenting to the emergency unit were transfused. Sickle cell disease (43%), acute bleeding (19.3%) and malaria (16.7%) were the leading diagnostic categories for which urgent blood transfusions were given.

Overall impression

The authors conducted a descriptive study, but methods of which lack details. There is a major concern for selection bias, since hemoglobin electrophoresis was only done among some patients, and yet sickle cell disease turned out to be the leading diagnostic category for transfusion. The data analysis seems incomplete (for the type of study design), since they could not even establish the factors associated with the main outcome variables. Consequently, the claims/answers to the current study are not supported by credible results. The authors make several speculations, such as about HIV, Schistosomiasis, and Sickle cell disease with no supporting data. Similarly, the results of some stated objectives are either not available or cancelled; probably for tests that were never performed. Results presentation and discussion deserve major revisions.

Specific issues, in details

Comment

Major issues

a) Introduction:

1. The objective(s) of the study, as stated in line 92-96, are not clear. Please revise, and clarify on the following:

i) Was the aim to ‘assess the rate’ of anemic children who….or to the ‘determine the proportion’ of anemic children who receive transfusion….?

ii) Did the authors examine (the indication, patterns and outcomes…), or describe?

Response

Objectives were revised, the word “assess” was replaced with “determine" and “describe” instead of “examine” in the specified lines as suggested. The word pattern was deleted from the objectives because it was not investigated. Please see lines 94- 97.

Comment

b) Methods:

1. The authors need to clarify on the type of study design (line 98). Was this a cross-sectional study?

Response

Cross-sectional study was added. Please see the method line 100. 

Comment

2. Generally, the methods lack details. By providing details of the scientific methods used in a clinical study, the researcher assures the global community that – another person, using the same tools may reproduce similar findings. Please explain and clarify on the following:

i) Samples of urine and stool were taken off (line 113); Provide details of the specific tests that were performed, and the machines used.

Response

Urine and stool analysis was conducted for schistosomal infection as described before. This has been inserted in the method. Please see the methods line 116.

Comment

ii) Line 114-115, mentions that blood was examined for malaria. Please provide details; which films: thick films, thin films, or both? Which stains were used? Who read them?

Response

Blood was examined for malaria using thick and thin blood films which were stained with 10% Giemsa and read by expert microscopist. This point has been inserted in the method. Please see it in lines 117-118.

Comment

iii) The diagnosis of anemia is very crucial in this study (refer to line 116). How was hemoglobin determined? Please provide details (e.g which machine was used?)

Response

 Hemoglobin estimation was done as part of the complete hemogram via automated blood-cell analyzer machine (Sysmex Hematology Analyzer; Sysmex, Kakogawa, Japan). This point has been inserted in the method as suggested. Please see lines 118-120.

Comment

iv) Hemoglobin electrophoresis (refer to line 116). Provide details of the machine and techniques used.

v) Hemoglobin electrophoresis (refer to line 116). The authors say this was done “if required in some cases”! This is a major concern for selection bias, if such a test that confirms sickle cell status was only performed among some patients. Remarkably sickle cell disease turned out to be the leading diagnostic category for transfusion, yet some children were NOT tested! Please explain.

Response

The test was performed for those who were suspected clinically to have sickle cell disease; the test was not done to already diagnosed patients, we depended on the recorded results from past tests. Haemoglobin genotype using the usual electrophoretic method (electrophoretic equipment model MUPID-EXU, Japan). Please see lines 127-131.

Comment

vi) In line 123, the authors state that transfusions were given for longer than 4 hours. However, transfusions are generally given for a period 2-4hrs. The practice of transfusion for longer than 4hrs is not safe and increases the risk for bacterial contamination. Is this the standard practice at Gadarif hospital? Please Explain.

Response

Yes it is correct that , transfusions are generally given for a period 2-4hrs. It was typo error (we mean for not longer than) it has been corrected. Please see the text line 138

Comment

vii) Similarly, explain how the diagnosis /suspicion of pulmonary edema was made. There is a risk for irrational use of frusemide.

Response

Pulmonary edema was clinically determined if the patient had shortness of breathing, increased respiratory rate with coarse crackles at the lung bases and radiologically as defined by the presence of Kerley B lines in the anteroposterior chest viewwas explained in the text. This point has been inserted in the method lines 140-143.

Comment

viii) The text in line 126 is confusing. Do you mean “the time taken from order, to actual transfusion”? This may not be referred to as ‘time of blood transfusion’. Please modify.

Response

The sentence was modified to “The time taken from order of blood to actual transfusion” Please see it now line 151.

Comment

ix) There are many other details missing, that should be mentioned such as:

x) How was the diagnosis of acute malnutrition made? Provide details.

Response

Details of diagnosis of malnutrition were added in the methods “Severe acute malnutrition was diagnosed following the WHO guidelines for malnutrition. A child was considered as severe acute malnutrition if weight-for-height z score was <-2 SD for age and sex or presence of bilateral lower limb edema”. Please see it lines 143-146.

Comment

xi) How was the diagnosis of visceral leishmaniasis made? Provide details.

Response

Details of diagnosis of leishmaniasis were added “The diagnosis of visceral leishmaniasis was confirmed by the visualization of the amastigote form of the parasite by microscopic examination of aspirates from or bone marrow using Giemsa-stain”. Lines 146-148.

Comment

xii) How was the diagnosis of sepsis made?

Response

Diagnosis of sepsis was explained in the text” The definition of sepsis is considered as “life threatening organ dysfunction caused by a dysregulated host response to infection[25]. Lines 149-150.

Comment

3. Statistical analysis:

i) Generally, additional statistical analysis needs to be done, such as univariate and bivariable analysis - to establish factors associated with the main outcomes such as survival, etc.

Response

We agreed that univariate and bivariable analysis is needed. However, there are few numbers “ 9 children discharged against medical advice and 6 children died” in the outcome. If we tried to conduct bivariable analysis in this case the model will be distorted with a wide range of confidence intervals. Hence this is the reason behind that we did not conduct it.

Comment

ii) Line 145: were all the numerical data symmetrical, to justify the use of Mean (SD)? If not, use median for asymmetrical data. Please check and revise.

Response

Yes Continuous data were checked for normality using Shapiro-Wilk test and they were presented as mean (Standard Deviation – SD) if they were normally distributed or median [interquartile range – IQR] if they were not normally distributed. This point has been inserted in the statistics. Please see lines 173-175.

Comment

4. For this study, the authors need to specify their operational definitions. For example;

i) What defined malaria?

ii) What defined sickle cell disease?

iii) What defined acute malnutrition?

iv) What defined acute bleeding?..etc.

to add the other definitions

Response

All these have been defined as suggested. Please see the methods and mentioned above.

Comment

c) Results:

1. Generally when presenting results, Table 1; which is the table of baseline characteristics comes up-front. These characteristics may include; gender, age, duration of illness, relationship with primary caregiver, occupation of caregiver, ethnic group/tribe etc. This table is missing.

Response

The suggested table has been inserted. Thank you very much indeed as you draw its outlines. 

Comment

2. Present the results systematically; objective by objective.

Systematic presentations of the results

3. Results of some stated objectives are missing, such as; ‘pattern’ and ‘time taken from order, to actual transfusion’. Provide them.

Results of pattern, time taken from order to actual transfusion.

Response

The pattern was not investigated and it has been deleted from the objectives. The others have been put in order as suggested. 

Comment

4. Samples of urine and stool were taken off. Provide the results of these tests.

Response

It has been added” There was no case of intestinal or urinary schistosomiasis” Please see line 237

Comment

5. Hemoglobin electrophoresis was done. Provide detailed results, including the proportion with SS, AS etc.

Response

The details have been inserted in the results. Please see lines 236.

Comment

6. Figure 1(line 154-166): The flow chart:

Consider revising this flow chart, such as below (only a guide);…Chart is attached, separate file,

Response

The suggested revised chart has been added. Thank very much indeed for this suggestion.

Comment

7. The authors should account for each study participant. For example, 141/300 transfused children were categorized as severe anemia. What was the diagnosis among the 159 with Hb>5 but got transfused? Please explain and modify.

Response

Yes the details of blood transfusion in this group as” Of 204 children who had hemoglobin levels of 4–7 g/dl and were transfused; 110 (53.9%) had heart failure, 38 (18.6%) had < 20% of red blood cells parasitized by malaria parasites, 32 (15.7%) had continuing bleeding, 9(4.4%) had combined/ others reasons, 8(3.9%) had dehydration and 7(3.4%) had impaired level of consciousness” Please see lines 240-244.

Comment

8. These 159, deserve a thorough discussion (in the discussion section).

Add to discussion Hb>5 and transfused

Response

Please see above

Comment

9. This account above is important, because the description offered in line, 170 -173; does not seem to tie-up with what we normally see clinically: That 91% had pallor, 87% with fever and 64% with breathlessness, yet only 141 (47%) were diagnosed as severe anemia. Please clarify.

To explain why only 47% had severe anemia where 91% had pallor & 87% had fever

Response

Yes we agreed it is a valid point. The explanation for this point is that symptoms might not be a useful indicator for anemia. This point has been inserted in the discussion. Please see line 302-305.

Comment

10. Line 181: Figure 2: This is a duplication of presentation of findings. Generally it is not necessary to show both the text (177-180) and figure. You choose either.

Figure 2 to be deleted or otherwise choose the figure and sentence below

If choose text, then one writes...’the diagnosis associated with order of blood transfusion is shown in figure 2, below:’

Response

Oaky the text was removed and the figure is retained. 

Comment

11. Line 189: For the five that developed adverse events, provide more details. E.g. After stopping, what happened to them? Full recovery? Where these 5 among the the 6 deaths? or the 9 that discharged self..

Response

The details have been inserted. Please see 248-250.

12. Fully account for each participant; such as in # c(11), and c(6) - c(8) above.

Response

Please see above

Comment

13. Line 190: The phrase; “A total of 285 children improved…” is left hanging. So what factors were associated with recovery? This entire paragraph (190-194) does not make sense, without additional statistical analysis, as pointed out in c(3) above. Please revise.

factors associated with recovery needed?????

Response 

We agreed that there should be more analysis on the factors and their outcomes. As we mentioned above 285 out three hundreds improved. This left only 15 (died and discharged against medical advice) which is small number to perform bivariate analysis 

Comment

d) Discussion:

1. What is the burden of Sickle cell disease in Sudan? In line 236, they mention one in 123 children. Do the authors believe this figure/prevalence? ….especially in light of their current finding that show; 129/300 (43%) of transfused children in that same setting have sickle cell disease? Please explain, and discuss.

Response

The (Misseriya tribes) ethnicity and the consanguineous marriages were the main explanations of the high prevalence (24.9%) of carriers of HbS allele (HbAS) in certain areas of Sudan. This point has been added to the discussion as suggested. Please see lines 308-310.

Comment

2. This section is generally poorly written (flow), with lots of repetitions. Suggested flow may include the following areas (paragraphs):

i) First paragraph – what was the aim of the study/ what was your question?

ii) What did you find (the answer to your question)

iii) A brief explanation of the findings, in comparison/contrast with other studies.

iv) What is new in the current study?

v) The strength of your study.

vi) Limitations

vii) Conclusions /or and recommendations.

Response

All these points have been addressed.

e) Conclusion:

Generally, the conclusion shall need to be revised, once most of the above revisions are finalized, especially those related to additional statistical analysis.

Response

All these have been addressed in the discussion as suggested.

Minor issues

Comment

a) Abstract:

1. Were all the 1,800 children evaluated for anemia? Please clarify.

Response

Yes were assessed for anemia. This point has been inserted in the abstract as suggested.

Comment

2. The text in line, 46-47 is confusion. “The diagnoses associated with the order for anemia…” Do clinicians order for anemia? Consider revising text, such as; “The diagnoses associated with the order for blood transfusion were…..” Do the same elsewhere, such as line 177.

Response

The sentence now read (diagnoses associated with the “severity” instead of “order” of anemia). It has been corrected throughout the paper as suggested

Comment

b) Introduction:

1. The text in line 72-73 lack clarity. Consider revising text, such as; ‘Childhood anemia is a major cause of morbidity and mortality globally.

Response

The sentence was rephrased

Comment

c) Methods:

1. Line 99, add ‘city’ to ‘Gadarif’ so that text reads; Gadarif city is 400km…..

Response

The word city is added now

Comment

2. Sample size estimation: Specify the formula used; such as Kish, Leslie for cross-sectional studies.

Response

Formula specification has been inserted as suggested.

Comment

d) Results,

1. Line 170: The authors need to use terminologies correctly and in context. By ‘breathlessness’ do the authors mean, respiratory distress or difficulty in breathing? Please clarify.

Response

Breathlessness explained as difficulty in breathing.

Comment

2. Similarly, Line 171: Palpitation is a rare symptom among children, especially in <5 years who often don't report symptoms. Remember 151 (50.3%) were <5 years! In addition, in table 1; palpitation is grouped as a sign! This is confusing. Please revise.

Response

I agreed it is confusing “ palpitation” has been deleted from the text and the table.

Commet

3. Line 184: revise the phrase, such as; Median time, from order to time of blood issue (or better start of transfusion) – as appropriate.

Response

The sentence was rephrased as: The median (interquartile range) request-to-issue time for blood was 6 (3−12) hours.

Comment

4. Line 186: For fever, generally we consider an increment of >1°C, from baseline. Was this the case? Please clarify.

Response

The fever was defined.

Comment

5. Line 187 (Adverse events are very important): Better specify; of the 5, how many got fever, and how many got urticarial rash?

Response

The details have been inserted as all of them had fever and urticarial rash. Please see lines 247-250.

Comment

e) Discussion:

1. Line 201: The authors state that, “the prevalence of anemia in our study was slightly higher than…”. Why? Please discuss. But, first clarify on how the 531(diagnosed with anemia) were got from the 1,800...to exclude selection bias. In other words, were all the 1,800 admitted children evaluated for anemia? Please clarify.

Response

This was clarified in the results section, please see above.

Comment

2. The phrase in line 215 conveys no meaning, especially with regard to what is put in parenthesis or (....). Please revise.

Response

Sentence rephrased

Comment

3. Line 216-217, about schistosomiasis: Are you sure? Please provide results of stool and urine, showing how many did not have. These results are missing.

Response

The sentence is rephrased as “Shistosomal infection was reported as the main cause of severe anemia of children….” This fact was cited from a previous study and not as a finding of this study.

Comment

4. Line 218-219; how did you know? When HIV was not tested. No mention of HIV testing in the methods or results. This should be one of the limitation of the study

Response

This part was deleted as you can see. The HIV was not tested, this has been inserted in the limitation of the study. Please see lines 341.

Comment

5. Line 252-253: This phrase is not accurate; 0.4% in Kiguli et al study is not comparable to 1.6%! Please revise or explain.

Response

The sentence was rephrased as follows: The “probable” blood transfusion reaction in our study was higher than that reported by Kiguli et al

f) Other comments: None.

Reviewer #2: PLONE-D-19-15943

Title

Severe childhood anemia and emergency blood transfusion in Gadarif Hospital, eastern Sudan

GENERAL COMMENTS

In this study the Authors assessed the prevalence of anemia in children admitted to their hospital in Sudan with an indication for blood transfusion through an eight months period. We subsequently examined a series of pre- and post- transfusion parameters (underlying disease, symptoms, signs, patients’ outcome and others).

The study, as the first performed in Sudan in this field, appears original and useful to better understand the real prevalence of anemia, its causes, the indications to blood transfusion, the transfusion behavior in a pediatric emergency department and the patients’ outcome.

However, I have some remarks and suggestions for changes.

Comment

First of all, I think that a clear statement (with an appropriate reference) must be included with respect to the definition of anemia used by the Authors, because this can affect the results.

According to WHO, a child between 6 and 59 months is defined as “anemic” when the hemoglobin level is below 11.5 g/dl (“severely anemic” below 7 g/dl). Between 5 and 11 years “anemia” is present with Hb below 11.0 g/dl; between 12 and 14 years below 12.0 g/dl and from 15 years on below 13.0 g/dl. Except for the first age class, the definition of “severe anemia” is restricted to patients with hemoglobin level below 8.0 g/dl.

See: World Health Organization. Haemoglobin concentrations for the diagnosis of anaemia and assessment of severity. WHO; 2011.

Has to define all conditions in a paragraph titled operational definitions

Response

Yes thank you very much. The provided definition for anemia has been inserted as well the reference provided. Please see lines 121-126.

Comment

In the Results section of the abstract (line 42) the sentence was: “…513 (28.5%) were anemic (hemoglobin < 11 g/dl) and 141 (7.8%) had severe anemia” (hemoglobin level < 5 g/dl).” Please adjust this point.

Response

Okay thank you very much for your suggestion. The point has been adjusted. 

Comment

In the statistical section the Authors declare that data are presented as mean (with standard deviation), but in line 184-185 they cite a “median (interquartile range)”. In line 194 the haemoglobin level should be expressed as median [IQR], especially since the word in brackets is “range”. Median and IQR are also cited in lines 242 and 244.

Clarification of mean interquartile usage in an expression of mean SD 

In my opinion the sentence in statistical section should be “Data are presented as mean (± Standard Deviation – SD) or median [interquartile range – IQR]

Response

Yes we agreed and this has been adjusted all over the paper.

Comment

 In the same sections the Authors should be described the methodology used to calculate the sample size (see line 132 – 135)

Response 

Yes the sample size has been clarified. Please see lines 159-160

OTHER SPECIFIC COMMENTS

Introduction, Line 72 – 88

I would suggest a complete revision of the text, to make it more concise. The contents are appropriate, but there is a series of repetitions and redundancies.

Response

Repetitions and redundancies removed

Comment

Line 85

The reference n.12 (“Guidelines for informing about clinical decisions on transfusion associated with hemoglobin levels” ) doesn’t correspond to the WHO’s guidelines. In my opinion it could be better the reference n.20.

Response 

Yes we agreed and it has been replaced by the suggested reference. 

Reference 20 considered to replace reference 12

Line 123

Furosemide instead of frusemide

Response

Yes corrected as suggested

Comment

Line 135

A difference of 5% of what? Please specify.

Response

It is the precision (statistical term), it has been corrected.

Comment 

Line 148 (Patients’ characteristics)

It is the only subheading in the “Results” section: the Authors may choose to erase it, to add other subheadings, if deemed appropriate.

Response

The subheading was deleted as suggested.

Comment

Line 151

The question is always the same: is the definition of severe anemia arbitrarily chosen or reference-based? It is sufficient to declare it.

Response

The definition of anemia is included as suggested 

Comment

Figure 1. and lines 167-169

I suggest a table instead of a flowchart with the subsequent descriptive text.

The table is in the attachment named as “Suggested Table”

Response 

The suggested table added

Comment 

I would appreciate a bar chart with hemoglobin level classes at admission (e.g. < 4 / 4 - <5 / 5 - < 6 and so on) in the x axis and the prevalence in the y axis.

A bar chart to be added

Response

Oaky the requested figure has been inserted, please see figure 2.

Comment

Line 194

See “statistic section” in the General comments

Response

Statistics revised

Comment

Discussion

Line 202

I suggest to add the following reference: Kassebaum NJ, Jasrasaria R, Naghavi M, Wulf SK, Johns N, et al. A systematic analysis of global anemia burden from 1990 to 2010. Blood 2014(123):615–24. https://doi.org/10.1182/blood-2013-06-508325

Response

The reference is added

Comment

Line 251

I suggest to add also the following reference and include a short discussion on its different results respect to the articles in reference 2 and 11. Maitland K, Kiguli S, Olupot-Olupot P, Engoru C, Mallewa M, Saramago Goncalves P, et al. Immediate Transfusion in African Children with Uncomplicated Severe Anemia. New England Journal of Medicine. 2019 Aug 1;381(5):407–19.

Response

The reference is added, please see line

Comment 

Conclusion

“Future reach on anemia is required” The phrase is not clear: please reword.

Response

It has been changed

Comment

Lastly, it could be interesting to know the level of blood transfusion appropriateness. Are the WHO guidelines always followed? Did the Authors document cases of undertransfusion or non-transfusion in patients in need for homologous blood supply?

To add the information of adherence to WHO guidelines on blood transfusion.

Response

 It is a valid point and thank you very much. However we did not think of it in our objectives. Perhaps we will include it in our future research.

Regards

---

## [Decision Letter · Decision Letter 1]

30 Oct 2019

PONE-D-19-15934R1

Severe childhood anemia and emergency blood transfusion in Gadarif Hospital, eastern Sudan

PLOS ONE

Dear Professor Adam,

Thank you for submitting your manuscript to PLOS ONE. After careful consideration, we feel that it has merit but does not fully meet PLOS ONE’s publication criteria as it currently stands. Therefore, we invite you to submit a revised version of the manuscript that addresses the points raised during the review process.

Unfortunately we only received feedback on your revised manuscript from one reviewer. If the other one comes in very soon, I will send you his/her comments. Otherwise proceed with a second revision taking into account the comments of the reviewer. Check your manuscrit  for redundancies and replace Figure 1  by the new Table 1.

We would appreciate receiving your revised manuscript by Dec 14 2019 11:59PM. To enhance the reproducibility of your results, we recommend that if applicable you deposit your laboratory protocols in protocols.io, where a protocol can be assigned its own identifier (DOI) such that it can be cited independently in the future. For instructions see: http://journals.plos.org/plosone/s/submission-guidelines#loc-laboratory-protocols

A rebuttal letter that responds to each point raised by the academic editor and reviewer(s). This letter should be uploaded as separate file and labeled 'Response to Reviewers'.A marked-up copy of your manuscript that highlights changes made to the original version. This file should be uploaded as separate file and labeled 'Revised Manuscript with Track Changes'An unmarked version of your revised paper without tracked changes. This file should be uploaded as separate file and labeled 'Manuscript'.

We look forward to receiving your revised manuscript.

Kind regards,

Henk D. F. H. Schallig, Ph.D

Academic Editor

PLOS ONE

Reviewers' comments:

Reviewer's Responses to Questions

**Comments to the Author**

1. If the authors have adequately addressed your comments raised in a previous round of review and you feel that this manuscript is now acceptable for publication, you may indicate that here to bypass the “Comments to the Author” section, enter your conflict of interest statement in the “Confidential to Editor” section, and submit your "Accept" recommendation.

Reviewer #2: (No Response)

2. Is the manuscript technically sound, and do the data support the conclusions?

Reviewer #2: Yes

3. Has the statistical analysis been performed appropriately and rigorously? 

Reviewer #2: Yes

4. Have the authors made all data underlying the findings in their manuscript fully available?

Reviewer #2: Yes

5. Is the manuscript presented in an intelligible fashion and written in standard English?

Reviewer #2: Yes

6. Review Comments to the Author

Reviewer #2: All the remarks are amended, except the redundancies in the Introduction section.

When I said "redundancies", I meant: "children" two times in the first line and "mortality in children with severe anemia" two times in line 81 and 82. Please, arrange the text in a different fashion.

Figure 1 can be totally replaced by the new Table 1.

7. PLOS authors have the option to publish the peer review history of their article (what does this mean?). If published, this will include your full peer review and any attached files.

Reviewer #2: No

---

## [Author Response · Author response to Decision Letter 1]

31 Oct 2019

We would like to thank the editor and reviewers for their valuable comments on this manuscript. We greatly appreciate your time dealing with the manuscript and assessing the review comments, which enabled us to greatly improve the quality of our manuscript. Here is a point-by-point response to the reviewers’ comments and concerns

Comment

reviewer #2: All the remarks are amended, except the redundancies in the Introduction section.

When I said "redundancies", I meant: "children" two times in the first line and "mortality in children with severe anemia" two times in line 81 and 82. Please, arrange the text in a different fashion.

Figure 1 can be totally replaced by the new Table 1

Response

The redundancies in the Introduction section have been removed as suggested.

Please see the introduction now.

Figure 1 has been deleted as suggested.

Regards

---

## [Editor Report · Decision Letter 2]

12 Nov 2019

Severe childhood anemia and emergency blood transfusion in Gadarif Hospital, eastern Sudan

PONE-D-19-15934R2

Dear Dr. Adam,

We are pleased to inform you that your manuscript has been judged scientifically suitable for publication and will be formally accepted for publication once it complies with all outstanding technical requirements.

With kind regards,

Henk D. F. H. Schallig, Ph.D

Academic Editor

PLOS ONE
---

## [Editor Report · Acceptance letter]

19 Nov 2019

PONE-D-19-15934R2 

Severe childhood anemia and emergency blood transfusion in Gadarif Hospital, eastern Sudan 

Dear Dr. Adam:

I am pleased to inform you that your manuscript has been deemed suitable for publication in PLOS ONE. Congratulations! Your manuscript is now with our production department. 

With kind regards,

on behalf of

Dr. Henk D. F. H. Schallig 

Academic Editor

PLOS ONE